# Hierarchical Catalogue Generation for Literature Review: A Benchmark

**Kun Zhu**[†]**, Xiaocheng Feng**[†‡]**, Xiachong Feng**[†]**, Yingsheng Wu**[†]**, Bing Qin**[†‡]

[†]Harbin Institute of Technology     [‡] Peng Cheng Laboratory

{kzhu,xcfeng,xiachongfeng,yswu,qinb}@ir.hit.edu.cn

## Abstract

Scientific literature review generation aims to extract and organize important information from an abundant collection of reference papers and produces corresponding reviews while lacking a clear and logical hierarchy. We observe that a high-quality catalogue-guided generation process can effectively alleviate this problem. Therefore, we present an atomic and challenging task named Hierarchical Catalogue Generation for Literature Review as the first step for review generation, which aims to produce a hierarchical catalogue of a review paper given various references. We construct a novel English Hierarchical Catalogues of Literature Reviews Dataset with 7.6k literature review catalogues and 389k reference papers. To accurately assess the model performance, we design two evaluation metrics for informativeness and similarity to ground truth from semantics and structure. Our extensive analyses verify the high quality of our dataset and the effectiveness of our evaluation metrics. We further benchmark diverse experiments on state-of-the-art summarization models like BART and large language models like ChatGPT to evaluate their capabilities. We further discuss potential directions for this task to motivate future research.

## 1 Introduction

Today's researchers can publish their work not only in traditional venues like conferences and journals but also in e-preprint libraries and mega-journals, which is fast, convenient, and easy to access (Fire and Guestrin, 2019). This enables rapid development and sharing of academic achievements. For example, statistics show that more than 50,000 publications occurred in 2020 in response to COVID-19 (Wang and Lo, 2021). Therefore, researchers are overwhelmed by considerable reading with the explosive growth in the number of scientific papers, urging more focus on scientific literature review generation (Altmami and Menai, 2022).

Pioneer studies on scientific literature review generation explore citation sentence generation (Xing et al., 2020; Luu et al., 2020; Ge et al., 2021; Wu et al., 2021) and related work generation (Hoang and Kan, 2010; Hu and Wan, 2014; Li et al., 2022; Wang et al., 2022). However, these methods can only generate short summaries, while a literature review is required to provide a comprehensive and sufficient overview of a particular topic (Webster and Watson, 2002). Benefiting from the development of language modeling (Lewis et al., 2019), recent works directly attempt survey generation (Mohammad et al., 2009; Jha et al., 2015; Shuaiqi et al., 2022) but usually suffer from a disorganized generation without hierarchical guidance. As illustrated in Figure 1(A), we take gpt-3.5-turbo[1], a large language model trained on massive amounts of diverse data including scientific papers, as the summarizer to generate scientific literature reviews. Direct generation can lead to disorganized reviews with content repetition and logical confusion.

Since hierarchical guidance is effective for text generation (Yao et al., 2019), as shown in Figure 1(B), we observe that the catalogue, representing the author's understanding and organization of existing research, is also beneficial for scientific literature review generation. Therefore, the generation process can be divided into two steps. First is generating a hierarchical catalogue, and next is each part of the review. Figure 1(C) reveals that even state-of-the-art language models can not obtain a reliable catalogue, leaving a valuable and challenging problem for scientific literature review generation.

To enable the capability of generating reasonable hierarchical catalogues, we propose a novel and challenging task of **Hi**erarchical **Cat**alogue **G**eneration for scientific **L**iterature **R**eview, named as **HiCatGLR**, which is a first step towards automatic review generation. We construct the first benchmark for HiCatGLR by gathering *html* for-

---

[1]A version of ChatGPT. openai.com/blog/chatgpt

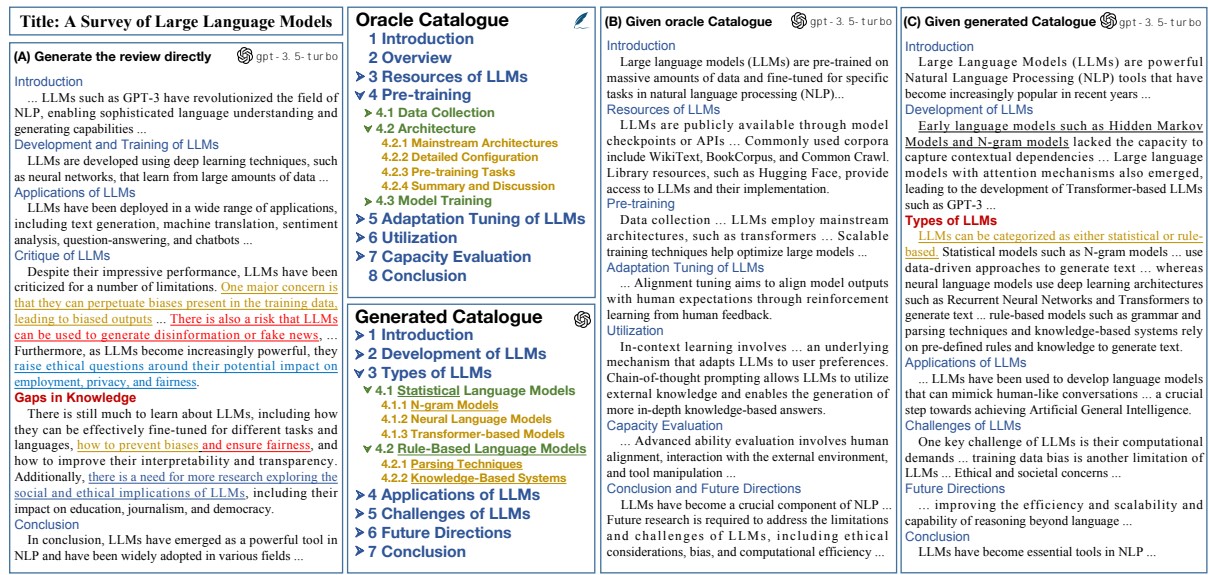

Figure 1: Examples of scientific literature review generated by gpt-3.5-turbo, given the title "A Survey of Large Language Models." (A) represents direct generation with the title only. Duplicated contents (underlined) exist in two different chapters without proper hierarchical guidance. Besides, the title "Gaps in Knowledge" should be part of "Critique of LLMs" rather than alongside it. (B) denotes the review based on both title and oracle (human-written) catalogue. Under the guidance of the hierarchical catalogue, the model can generate transparent and rational reviews. (C) consists of two steps: first generate a pseudo catalogue given the title and then obtain the entire review. It's apparent that the quality of generated catalogue is not satisfying since traditional language modeling methods, "statistical" and "rule-based," should have belonged to the chapter "Development of LLMs" rather than "Type of LLMs." This consequently causes the degeneration of reviews due to the unreliable catalogue.

mat[2] of survey papers' catalogues along with abstracts of their reference papers from Semantics Scholar[3]. After meticulous filtering and manual screening, we obtain the final **Hi**erarchical **Ca**talogue **D**ataset (**HiCaD**) with 7.6k references-catalogue pairs, which is the first to decompose the review generation process and seek to explicitly model a hierarchical catalogue. The resulting HiCaD has an average of 81.1 reference papers for each survey paper, resulting in an average input length of 21,548 (Table 1), along with carefully curated hierarchical catalogues as outputs.

Due to the structural nature of catalogues, traditional metrics like BLEU and ROUGE can not accurately reflect their generation quality. We specially design two novel evaluations for the catalogue generation, where Catalogue Edit Distance Similarity (CEDS) measures the similarity to ground truth and Catalogue Quality Estimate (CQE) measures the degree of catalogue standardization from the frequency of catalogue template words in results.

To evaluate the performance of various methods on our proposed HiCatGLR, we study both

end-to-end (one-step) generation and step-by-step generation for the hierarchical catalogues, where the former generally works better and the latter allows for more focus on target-level headings. We benchmark different methods under fine-tuned or zero-shot settings to observe their capabilities, including the recent large language models.

In summary, our contributions are threefold:

- We observe the significant effect of hierarchical guidance on literature review generation and propose a new task, HiCatGLR (Hierarchical Catalogue Generation for Literature Review), with the corresponding dataset HiCaD.

- We design several evaluation metrics for informativeness and structure accuracy of generated catalogues, whose effectiveness is ensured with detailed analyses.

- We study both fine-tuned and zero-shot settings to evaluate models' abilities on the HiCatGLR, including large language models.

## 2 Datasets: HICAD

We now introduce HiCaD, including the task definitions, data sources, and processing procedures. We also provide an overall statistical analysis.

---

[2] https://ar5iv.labs.arxiv.org/
[3] https://www.semanticscholar.org/

## 2.1 Definitions

The input of this task is the combination of the title $t$ of the target survey $S$ and representative information of reference articles $\{R^1, R^2, ..., R^n\}$, where $n$ is the number of reference articles cited by $S$. Considering the cost of data collection and experiments, we take abstracts as the representation of corresponding references. Besides, we restrict each abstract to 256 words, where the exceeding part will be truncated. The output is the catalogue $C = \{c^1, ..., c^k\}$ of $S$, where $k$ is the number of catalogue items. $c^i$ is an item in the catalogue consisting of a level mark $l^i \in \{L_1, L_2, L_3\}$, which represents the level of the catalogue item, and the content $\{w_1^i, w_2^i, ..., w_p^i\}$ with $p$ number of words. As shown in Figure 1, "*Pre-training*" is the first-level heading, "*Architecture*" is the second-level heading, and *" Mainstream Architectures"* is the third-level heading. In our experiment, we only keep up to the third-level headings and do not consider further lower-level headings.

## 2.2 Dataset Construction

Our dataset is collected from two sources: *arXiv*[4] and *Semantics Scholar*[5]. We keep papers[6] containing the words "survey" and "review" in the title and remove ones with "book review" and "comments". We finally select 11,435 papers that are considered to be review papers. It is straightforward to use a crawler to get all 11,435 papers in PDF format according to the *arxiv-id*. However, extracting catalogue from PDF files is difficult where structural information is usually dropped during converting. Therefore, we attempt *ar5iv*[7] to get the papers in HTML format. This website processes articles from *arXiv* as responsive HTML web pages by converting from LaTeX via *LaTeXML*[8]. Some authors do not upload their LaTeX code, we have to skip these and collect 8,397 papers.

For the output part, we obtain the original catalogues by cleaning up the HTML files. Then we replace the serial number from the heading with the level mark <$L_i$> using regex. For input, we collate the list of reference papers and only keep the valid papers where titles and abstracts exist. We convert

---

[4] https://arxiv.org/
[5] https://www.semanticscholar.org/
[6] These survey papers' metadata are obtained from Kaggle up to the end of April 2023. https://www.kaggle.com/datasets/Cornell-University/arxiv.
[7] https://ar5iv.org/
[8] https://github.com/brucemiller/LaTeXML

all words to lowercase for subsequent generation and evaluation. Finally, after removing data with less than 5 catalogue items and less than 10 valid references, we obtain 7,637 references-catalogue pairs. We count the fields to which each paper belongs (Table 6). We choose the computer science field with the largest number of papers for the experiment and split the 4,507 papers into training (80%), validation (10%), and test (10%) sets.

## 2.3 Dataset Statistics and Analysis

Taking the popular scientific dataset as an example, we present the characteristics of the different multi-document scientific summarization tasks in Table 1. Dataset Multi-Xscience is proposed by Lu et al. (2020), which focuses on writing the related work section of a paper based on its abstract with 4.4 articles cited in average. Dataset BigSurvey-MDS is the first large-scale multi-document scientific summarization dataset using review papers' introduction section as target (LIU et al., 2022), where previous work usually takes the section of related work as the target. Both BigSurvey and our HiCat-GLR task have more than 70 references, resulting in over 10,000 words of input, while their output is still the scale of a standard text paragraph, similar to Multi-Xscience. A natural difference between our task and others is that our output contains hierarchical structures, which place high demands on logic and conciseness for generation.

To measure how abstractive our target catalogues are, we present the proportion of novel n-grams in the target summaries that do not appear in the source (Table 2). The abstractiveness of HiCaD is lower than that of BigSurvey-MDS and Multi-XScience, which suggests that writing catalogues focus on extracting keywords from references. This conclusion is in line with the common sense that literature review is closer to reorganization than innovation. Therefore, our task especially challenges summarizing ability rather than generative ability.

We also analyze the share of each level of the catalogue in the whole catalogue. Figure 2 shows the value and proportional relationship of the average number of catalogue items as well as average word length at different levels. It can be seen that the second-level headings have the most weight, being 44.32% of the average number and 48.50% of the average word length. Table 3 shows the weight of the headings at each level in the catalogue from the perspective of word coverage. We calculate

| Datasets | Task | Pairs | Refs | #Sents (input) | #Words (input) | Form | #Sents (output) | #Words (output) |
|---|---|---|---|---|---|---|---|---|
| Multi-XScience | Related Work | 40,828 | 4.4 | - | 778.1 | Paragraph | - | 116.4 |
| BigSurvey-MDS | Survey Introduction | 4,478 | 76.3 | 450.1 | 11,893.1 | Paragraph | 38.8 | 1,051.7 |
| HiCaD | Survey Catalogue | 7,637 | 81.1 | 471.4 | 21,548.2 | Hierarchical Catalogue | 27.4 | 103.2 |

Table 1: Comparison of our HiCaD dataset to other multi-document scientific summarization tasks and their datasets. **Pairs** means the number of examples. **Refs** stands for the average number of input papers for each sample. **Sents** and **Words** indicate the average number of sentences and words in input or output calculated by concatenating all input or output sources. **Form** is the form of the output text.

| Datasets | % of novel n-grams in target summary | | | |
|---|---|---|---|---|
| | unigrams | bigrams | trigrams | 4-grams |
| Multi-XScience | 42.33 | 81.75 | 94.57 | 97.62 |
| BigSurvey-MDS | 37.39 | 76.46 | 93.87 | 98.04 |
| HiCaD | 23.13 | 64.66 | 87.11 | 95.22 |

Table 2: The proportion of novel n-grams in target summaries across different summarization datasets.

| | $L_1$ R-1/R-2/R-L | $L_2$ R-1/R-2/R-L | $L_3$ R-1/R-2/R-L |
|---|---|---|---|
| $L_1$ | / | 18.5/6.1/18.1 | 6.2/1.3/6.1 |
| $L_2$ | 18.5/6.1/18.1 | / | 11.0/3.2/10.9 |
| $L_3$ | 6.2/1.3/6.1 | 11.0/3.2/10.9 | / |
| **Total** | 45.3/39.1/45.3 | 57.9/53.0/57.9 | 25.1/23.0/25.1 |

Table 3: ROUGE scores between different levels of headings. **Total** represents the whole catalogue.

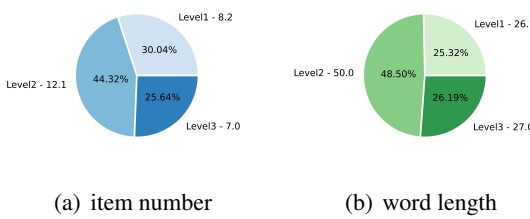

(a) item number      (b) word length

Figure 2: The value and proportional relationship of the average number of catalogue items as well as average word length at different levels.

ROUGE [9] scores between different levels of headings ($L_1$, $L_2$, $L_3$) and the general catalogue "Total". Similar to the above, the secondary headings have the highest Rouge-1 score of 57.9 for the entire catalogue. Moreover, the Rouge-1 score of 18.5 for the first and second-level headings $L_1$-$L_2$ indicates some overlaps between the first and second levels. The low Rouge scores of $L_1$-$L_3$ and $L_2$-$L_3$ reveal that there are indeed different usage and wording distributions between different levels.

## 3 Metrics

To evaluate the quality of generated catalogues, we propose two evaluation methods, Catalogue Quality Estimate (CQE) and Catalogue Edit Distance Similarity (CEDS). The former takes a textual perspective, while the latter integrates text and structure. We use these evaluation methods to assess the informativeness of the generated catalogues and the gap between the generated catalogues and the

---

oracle ones, respectively.

### 3.1 Catalogue Quality Estimate

There are some fixed templates in the catalogue, such as *introduction*, *methods*, and *conclusion*, which usually do not contain information about domain knowledge and references. The larger the percentage of template words in the catalog, the less valid information is available. Therefore, the proportion of template words can indicate the information content of the catalogue to some extent. We collate a list of templates and calculated the percentage of templates in the catalogue items as Catalogue Quality Estimate (CQE). CQE evaluation metric measures the informativeness of the generated catalogue through template words statistics. Table 7 lists all template words. The CQE of oracle catalogues in the test set is 11.1%.

### 3.2 Catalogue Edit Distance Similarity

Catalogue Edit Distance Similarity (CEDS) measures the semantic similarity and structural similarity between the generated and oracle catalogues. Traditional automatic evaluation methods in summarization tasks (such as ROUGE (Lin, 2004) and BERTScore (Zhang* et al., 2020)) can only measure semantic similarity. However, catalogues are texts with a hierarchical structure, so the level of headings in catalogues also matters.

We are inspired by the thought of edit distance which is commonly used in the ordered labeled tree similarity metric. An ordered labeled tree is one in which the order from left to right among siblings is significant (Zhang and Shasha, 1989). The tree

---

[9] The ROUGE scores in this paper are all computed by ROUGE-1.5.5 script with the option "-c 95 -r 1000 -n 2 -a -m"

edit distance (TED) between two ordered labelled trees $T_a, T_b$ is defined as the minimum number of node edit operations that transform $T_a$ into $T_b$ (Paaßen, 2018). There are three edit operations on the ordered labeled tree: deletion, insertion, and modification, each with a distance of one. Similar to TED, the definition of catalogue edit distance (CED) is the minimum distance of item edit operations that transform a catalogue $C_a$ into another catalogue $C_b$, where each entry in the catalogue is a node. The difference is that we calculate the distance between two items according to the similarity of nodes when modification:

$$\text{Distance}(x, y) = \min(1, \alpha \times (1 - \text{Similarity}(x, y)).$$

We leverage BERTScore to obtain the similarity between two items of the catalogue based on the embeddings from SciBERT (Beltagy et al., 2019):

$$\text{Similarity}(x, y) = (\text{Sci-})\text{BERTScore}(x, y).$$

SciBERT is more suitable for scientific literature than BERT because it was pretrained on a large multi-domain corpus of scientific publications. We take hyperparameter $\alpha = 1.2$ during experiments. Therefore, we can define Catalogue Edit Distance Similarity (CEDS) as:

$$\text{CEDS}(C_a, C_b) = 100 \times \left(1 - \frac{\text{CED}(C_a, C_b)}{\max(|C_a|, |C_b|)}\right).$$

We use the library *Python Edit Distances*[10] proposed by Paaßen et al. (2015) for the implementation of algorithms. A detailed example of node alignment and the conversion process between two catalogues is given in Appendix F.

## 4 Models

We now introduce our explorations in hierarchical catalogue generation for literature review, including end-to-end and step-by-step approaches.

### 4.1 End-to-End Approach

One of the main challenges in generating the catalogue is handling long input contexts. The intuitive and straightforward generation method is the end-to-end model, and we experiment with two models that specialize in processing long text. We consider an encoder-decoder model for the end-to-end approach. The model takes the title of a survey

[10] https://gitlab.ub.uni-bielefeld.de/bpaassen/python-edit-distances/-/tree/master

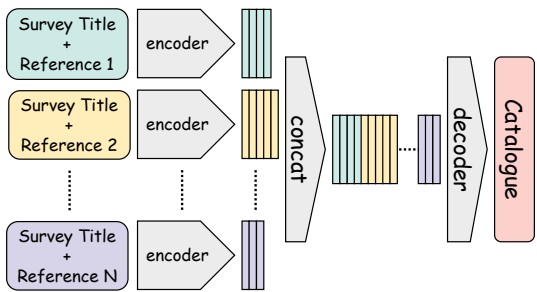

Figure 3: Architecture of the step-by-step approach.

and the information from its reference papers as input before generating the survey catalogue. The title and references are concatenated together and fed to the encoder. However, existing transformer-based models with $O(N^2)$ computational complexity show an explosion in computational overhead as the input length increases significantly.

We choose Fusion-in-Decoder (FiD) (Izacard and Grave, 2021), a framework specially designed for long context in open domain question answering, to handle the end-to-end catalogue generation. As shown in Figure 3, the framework processes the combination of the survey title and information from each reference paper independently by the encoder. The decoder pays attention to the concatenation of all representations from the encoder. Unlike previous encoder-decoder models, the FiD model processes papers separately in the encoder. Therefore, the input can be extended to a large number of contexts since it only performs self-attention over one reference paper each time. This allows the computation time of the model to grow linearly with the length of the input. Besides, the joint processing of reference papers in the decoder can better facilitate the interaction of multiple papers.

### 4.2 Step-by-Step Approach

Another main challenge in generating the catalogue is modeling relationships between catalogue items at different levels. Motivated by Tan et al. (2021), we explore an effective approach for incremental generation. Progressive generation divides the complicated problem of generating a complete catalogue into more manageable steps, namely the generation of hierarchical levels of the catalogue. Different from generating everything in one step, the progressive generation allows the model to perform high-level abstract planning and then shift attention to increasingly concrete details. Figure 5 illustrates the generation process.

| Method | | ROUGE-1/ROUGE-2/ROUGE-L ↑ | | | | BERTScore ↑ | CEDS↑ | CQE ↓ |
|---|---|---|---|---|---|---|---|---|
| | | $L_1$ | $L_2$ | $L_3$ | Total | | | |
| Extractive Methods | | | | | | | | |
| LexRank | | —— | —— | —— | 10.5/1.7/10.0 | 47.5 | —— | 2.4 |
| TextRank | | —— | —— | —— | 12.9/1.2/12.1 | 47.2 | —— | 2.1 |
| End-to-End | | | | | | | | |
| LED | base | 37.8/12.2/37.4‡ | 15.3/3.8/15.1 | 1.8/0.3/1.5 | 26.1/7.5/25.9 | 64.2 | 31.4† | 22.2 |
| | large | **39.3/12.8/38.7** | **18.9/4.0/18.8** | 5.2/1.0/5.2‡ | **29.5/8.0/29.3** | **64.7** | **34.1** | 16.9† |
| FiD(BART) | base | 35.1/10.3/34.8 | 15.1/3.8/15.0 | 3.0/0.4/2.9 | 25.1/7.1/25.0 | 64.3† | 29.5 | 21.4 |
| | large | 34.6/10.9/34.1 | 16.9/4.3/16.7† | **5.6/1.1/5.6** | 27.0/7.6/26.9‡ | 64.4‡ | 31.0 | 18.5 |
| Step-by-Step | | | | | | | | |
| LED | base | 36.3/12.0/36.0 | 13.0/2.9/12.9 | 0.7/0.1/0.7 | 24.2/6.7/24.0 | 63.0 | 30.1 | 23.0 |
| | large | 37.0/11.8/36.5 | 13.3/3.2/13.2 | 1.8/0.5/1.8 | 26.4/7.3/26.2 | 63.6 | 32.3‡ | 20.8 |
| FiD(BART) | base | 36.8/12.7/36.5 | 9.9/2.5/9.8 | 0.0/0.0/0.0 | 22.0/6.4/21.8 | 63.2 | 27.1 | 29.1 |
| | large | 37.7/11.8/37.2† | 8.8/2.1/8.7 | 0.6/0.0/0.6 | 24.3/6.4/24.0 | 63.3 | 28.9 | 21.6 |
| Large Language Models | | | | | | | | |
| Galactica-6.7b | | 16.0/4.7/15.5 | 2.0/0.3/1.9 | 0.8/0.0/0.7 | 17.7/4.3/17.3 | 55.6 | 28.0 | 13.8‡ |
| ChatGPT | | 34.1/11.0/33.4 | 18.2/3.9/18.0‡ | 4.8/0.8/4.8† | 26.6/6.8/26.3† | 62.6 | 28.0 | **13.2** |

Table 4: Automatic evaluation results on HiCaD. Bold indicates the best value in each setting. ‡ represents the second best result and † represents the third best.

## 5 Experiments

We study the performance of multiple models on the HiCaD dataset. Detailed analysis of the generation quality is provided, including correlation validation of the proposed evaluation metrics with human evaluation and ROUGE for abstractiveness.

## 5.1 Baselines

Due to the large input length, we choose an encoder-decoder transformer model that can handle the long text and its backbone model to implement FiD besides various extractive models in our experiments. (I) **LexRank** (Erkan and Radev, 2004) is an unsupervised extractive summarization approach based on graph-based centrality scoring of sentences. (II) **TextRank** (Mihalcea and Tarau, 2004) is a graph-based ranking algorithm improved from Google's PageRank (Page et al., 1999) for keyword extraction and document summarization, which uses co-occurrence information (semantics) between words within a document to extract keywords. (III) **BART** (Lewis et al., 2019) is a pre-trained sequence-to-sequence Transformer model to reconstruct the original input text from the corrupted text with a denoising auto-encoder. (IV) **Longformer-Encoder-Decoder** (LED) (Beltagy et al., 2020) is built on BART but adopts the sparse global attention mechanisms in the encoder part, which alleviates the input size limitations of the BART model (1024 tokens) to 16384 tokens.

## 5.2 Implementation Details

We use dropout with the probability $0.1$ and a learning rate of $4e - 5$. The optimizer is Adam with $\beta_1 = 0.9$ and $\beta_2 = 0.999$. We also adopt the learning rate warmup and decay. During the decoding process, we use beam search with a beam size of 4 and tri-gram blocking to reduce repetitions. We adopt the implementations of LED from HuggingFace's Transformers (Wolf et al., 2020) and FiD from SEGENC (Vig et al., 2022). We maximize the input length to 16,384 tokens. All the models are trained on one NVIDIA A100-PCIE-80GB.

## 5.3 Results

As shown in Table 4, we calculated the ROUGE scores, BERTScore, Catalogue Edit Distance Similarity (CEDS) and Catalogue Quality Estimate (CQE) between the generated catalogues and oracle ones separately. We especially remove all level mark symbols (e.g. $<L_1>$) for evaluation. In order to compare the ability of different methods to generate different levels of headings in detail, we calculated ROUGE scores for each level of headings with the corresponding level of oracle ones.

We first analyze the performance of traditional extractive approaches. Since these extractive models can not generate catalogues with an explicit hierarchy as abstractive methods, we only calculate the ROUGE scores of extracted results to the entire oracle catalogues (Column *Total* in Table 4). We take the ROUGE score (11.9/4.5/11.4) between titles of literature reviews and entire oracle catalogues

as a threshold because titles are the most concise and precise content related to reviews. LexRank (10.5/1.7/10.0) and TextRank (12.9/1.2/12.1) only achieves similarly results as the threshold. This means extractive methods are not suitable for hierarchical catalogue generation.

By comparing all evaluation scores, the end-to-end approach achieves higher similarity with the ground truth than step-by-step on the whole catalogue. Generally, the model with a larger number of parameters performs better. However, there are still duplication and hierarchical errors in the current results (See case studies in Appendix F).

Large language models have shown excellent performance on many downstream tasks due to massive and diverse pre-training. We also test two representative large language models: Galactica (Taylor et al., 2022) and ChatGPT (gpt-3.5-turbo). Galactica (GAL) is a large language model for science that achieve state-of-the-art results over many scientific tasks. The corpus it used includes over 48 million papers, textbooks, scientific websites and so on. ChatGPT is the best model recognized by the community and also trained on ultra large scale corpus. The corpus used by these two models in the pre-training phase contains scientific literature and thus we consider that the knowledge of the models contain the reference papers. From evaluation results, instructions understanding and answer readability, ChatGPT generates far better results than GAL. It's worth noting that large language models can not outperform models (LED-large) specially trained for this task. This reveals that simply stacking knowledge and parameters may not be a good solution for catalogue generation which requires more capabilities on logic and induction. See Appendix D for specific details and analysis.

### 5.4 Consistency Check

### 5.4.1 Human Evaluation

To demonstrate the validity of our proposed evaluation metrics, we conduct consistency tests on CEDS with human evaluation. We generate 50 catalogues for each implementation with a total number of 450 samples, each sample is evaluated by three professional evaluators. We skip Galactica-6.7b for subsequent experiments since it can hardly generate reasonable catalogues. Evaluators are required to assess the quality of catalogues based on the similarity (ranging from one to five, where five represents the most similar) to the oracle one.

First, we test the human evaluations and CEDS of corresponding data for normal distribution. After the Shapiro-Wilk test, the p-values for the two sets are $0.520$ and $0.250$, all greater than $0.05$ (Table 8). That means these data groups can be considered to meet the normal distribution, which enables further Pearson correlation analysis. Person correlation analysis shows that $p$-values between CEDS and Human are $0.027$, more diminutive than $0.05$ (Table 9). The $r$-value is $0.634$, which represents a strong positive correlation. Therefore, we consider CEDS as a valid evaluation indicator.

### 5.4.2 Automatic Evaluation

We also conduct a pair-wise consistency test between all the automatic indicators measured (Table 5) by Pearson's correlation coefficient, where only ROUGE-L in ROUGE was computed.

First, a noticeable trend is that the ROUGE-L of the first-level catalogue (L1RL) is not correlated with any other metrics. We infer that this is due to the ease of generating first-level headings, which perform similarly across methods and reach bottlenecks. The second & third-level catalogue (L2RL, L3RL) exhibit an extremely strong positive correlation with the total one (TotalRL), which suggests that the effectiveness of generating second & third-level headings affects the overall performance of catalogue generation. Second, Catalogue Quality Estimate (CQE) negatively correlates with the ROUGE-L in three levels (L2RL, L3RL, TotalRL). This indicates that domain knowledge and reference information are mainly found in the secondary and tertiary headings, which is in line with human perception. Finally, we study the difference between TotalRL and BERTScore. We find that TotalRL and BERTScore are not relevant, but they are correlated with CEDS respectively. This means that CEDS can combine both ROUGE and BERTScore indicators and better describe the similarity between generated results and oracle ones.

## 6  Related Work

**Survey generation** Our work belongs to the multi-document summarization task of generation, which aims to reduce the reading burden of researchers. Survey generation is the most challenging task in multi-document scientific summarization. The early work was mainly an extractive approach based on various content selection ways (Mohammad et al. (2009), Jha et al. (2015)). The results

|  |  | L1RL | L2RL | L3RL | TotalRL | BERTScore | CEDS | CQE |
|---|---|---|---|---|---|---|---|---|
| L1RL | r | 1 | -0.240 | -0.357 | 0.118 | 0.348 | 0.511 | 0.335 |
|  | p |  | 0.534 | 0.346 | 0.763 | 0.358 | 0.160 | 0.378 |
| L2RL | r | -0.240 | 1 | .889** | .847** | 0.424 | 0.511 | -.777* |
|  | p | 0.534 |  | 0.001 | 0.004 | 0.256 | 0.160 | 0.014 |
| L3RL | r | -0.357 | .889** | 1 | .826** | 0.434 | 0.409 | -.827** |
|  | p | 0.346 | 0.001 |  | 0.006 | 0.243 | 0.275 | 0.006 |
| TotalRL | r | 0.118 | .847** | .826** | 1 | 0.586 | .808** | -.790* |
|  | p | 0.763 | 0.004 | 0.006 |  | 0.098 | 0.008 | 0.011 |
| BERTScore | r | 0.348 | 0.424 | 0.434 | 0.586 | 1 | .707* | -0.082 |
|  | p | 0.358 | 0.256 | 0.243 | 0.098 |  | 0.033 | 0.834 |
| CEDS | r | 0.511 | 0.511 | 0.409 | .808** | .707* | 1 | -0.338 |
|  | p | 0.160 | 0.160 | 0.275 | 0.008 | 0.033 |  | 0.374 |
| CQE | r | 0.335 | -.777* | -.827** | -.790* | -0.082 | -0.338 | 1 |
|  | p | 0.378 | 0.014 | 0.006 | 0.011 | 0.834 | 0.374 |  |

Table 5: Results of Pearson's consistency test. ** represents a significant correlation at the 0.01 level (two-tailed). * denotes a significant correlation at the 0.05 level (two-tailed).

generated by unsupervised selection models have significant coherence and duplication problems.

To make the generated results have a clear hierarchy, Hoang and Kan (2010) additionally inputs an associated topic hierarchy tree that describes a target paper's topics to drive the creation of an extractive related work section. Sun and Zhuge (2019) proposes a template-based framework for survey paper automatic generation. It allows users to compose a template tree that consists of two types of nodes, dimension node and topic node. Shuaiqi et al. (2022) trains classifiers based on BERT(Devlin et al., 2018) to conduct category-based alignment, where each sentence from academic papers is annotated into five categories: background, objective, method, result, and other. Next, each research topic's sentences are summarized and concatenated together. However, these template trees are either inflexible or require the users to have some background knowledge to give, which could be more friendly to beginners. Also, this defeats the original purpose of automatic summarization to aid reading.

**Text structure Generation** There are some efforts involving the automatic generation of text structures. Xu et al. (2021) employs a hierarchical model to learn structure among paragraphs, selecting informative sentences to generate surveys. But this structure is not explicit. Tan et al. (2021) generate long text by producing domain-specific keywords and then refining them into complete passages. The keywords can be considered as a structure's guide for the subsequent generation. Trairatvorakul et al. reorganize the hierarchical structures of three to six articles to generate a hierarchical structure for a survey. Fu et al. (2022) gen-

erate slides from a multi-modal document. They generate slide titles, i.e., structures of one target document, via summarizing and placing an object. These efforts don't really model the relationships between multiple documents, which is far from the hierarchical structure of a review paper. This paper focuses on the generation of a hierarchical catalogue for literature review. Traditional automatic evaluation methods for summarization calculates similarity among unstructured texts from the perspective of overlapping units and contextual word embeddings (Papineni et al. (2002), Banerjee and Lavie (2005), Lin (2004), Zhang* et al. (2020), Zhao et al. (2019)). Tasks involving structured text generation are mostly measured using the accuracy, such as SQL statement generation (He et al. (2019), Wang et al. (2019)), which ignores semantic information. Assessing the similarity of catalogues to standard answers should consider both structural and semantic information.

## 7 Conclusion

In this work, we observe the significant effect of hierarchical guidance on literature review generation and introduce a new task called hierarchical catalogue generation for literature review (HiCatGLR) and develop a dataset HiCaD out of arXiv and Semantic Scholar Dataset. We empirically evaluate two methods with eight implementations for catalogue generation and found that the end-to-end model based on LED-large achieves the best result. CQE and CEDS are designed to measure the quality of generated results from informativeness, semantics, and hierarchical structures. Dataset and code are available at https://github.com/zhukun1020/HiCaD. Our analy-

sis illustrates how CEDS resolves some of the limitations of current metrics. In the future, we plan to explore a better understanding of input content and the solution to headings level error, repetition, and semantic conflict between headings.

## Limitations

The hierarchical catalogue generation task can help with literature review generation. However, there is still much room for improvement in the current stage, especially in generating the second & third-level headings, which requires a comprehensive understanding of all corresponding references. Besides, models will need to understand each reference completely rather than just their abstracts in the future. Our dataset, HiCaD, does not cover a comprehensive range of domains. In the future, we need to continue to expand the review data in other fields, such as medicine. Currently, we only experiment on a single domain due to limitations of collected data resources, where knowledge transfer across domains matters for catalogues generation.

## Ethics Statement

In this paper, we present a new dataset, and we discuss here some relevant ethical considerations. (1) Intellectual property. The survey papers and their reference papers used for dataset construction are shared under the CC BY-SA 4.0 license. It is free for research use. (2) Treatment of data annotators. We hire the annotators from proper channels and pay them fairly for the agreed-upon salaries and workloads. All hiring is done under contract and in accordance with local regulations.

## Acknowledgements

Xiaocheng Feng is the corresponding author of this work. We thank the anonymous reviewers for their insightful comments. This work was supported by the National Key R&D Program of China via grant No.2020AAA0106502, National Natural Science Foundation of China (NSFC) via grant 62276078, the Key R&D Program of Heilongjiang via grant 2022ZX01A32, the International Cooperation Project of PCL, PCL2022D01 and the Fundamental Research Funds for the Central Universities (Grant No.HIT.OCEF.2023018).

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

## A  Dataset Domain

We count the fields to which the dataset belongs, and the computer science field have the most papers, followed by mathematics. In this paper, we use only papers from the computer science domain for experiments.

| Domain | Subject | Amount |
|---|---|---|
| Computer Science | Computing Research Repository | 4,507 |
| Mathematics | Mathematics | 922 |
| Physics | Physics | 284 |
| | High Energy Physics - Phenomenology | 235 |
| | Condensed Matter | 233 |
| | High Energy Physics - Theory | 173 |
| | Quantum Physics | 157 |
| | General Relativity and Quantum Cosmology | 103 |
| | High Energy Physics - Experiment | 90 |
| | Mathematical Physics | 76 |
| | Nuclear Experiment | 59 |
| | Nuclear Theory | 45 |
| | Nonlinear Sciences | 27 |
| | High Energy Physics - Lattice | 23 |
| Electrical Engineering & Systems Science | Electrical Engineering and Systems Science | 305 |
| Statistics | Statistics | 267 |
| Quantitative Biology | Quantitative Biology | 83 |
| Quantitative Finance | Quantitative Finance | 33 |
| Economics | Economics | 15 |

Table 6: Domain distribution of literature review papers in the dataset HiCaD. Category division comes from arXiv.org.

## B  Template Words

The list of template words used when calculating CQE in this paper is listed in Table 7.

## C  Consistency Check

## D  Large Language Models

In this section, we experiment with the effect of two foundation models on the hierarchical catalogue generation task.

```
introduction    preliminaries    preliminary
background    overview    definition(s)    methodology
method(s)    sources    purpose    related    work    data
dataset(s)    outlook    eassy    benefit(s)    advantage(s)
disadvantage(s)    challenge(s)    application(s)
future    summary    notation(s)    result(s)    discussion
analysis    observation(s)    conclusion(s)    references
```

Table 7: All template words used to calculate Catalogue Quality Estimate (CQE).

| | Shapiro-Wilk test |
|---|---|
| **Human**$_{sample}$ | 0.250 |
| **CEDS**$_{sample}$ | 0.027 |

Table 8: P-values of Shapiro-Wilk test for CEDS and human evaluation results.

### D.1 Galactica

Galactica has five sizes ranging from 125M to 120B parameters and the standard size 6.7B is the max size we can use. We experimented one-shot test on our test set. There are two generation samples in Figure 4. It can be seen that there is a lot of repetition in the results.

### D.2 ChatGPT

The prompt of what we use to generate a directory using ChatGPT is:

Your task is to write a table of contents for the review paper by recalling relevant papers, organizing and classifying them according to the given review paper topic. Only the first, second and third level headings need to be written, no detailed explanation is required. Please ensure that your catalogue is well-structured, clear, and concise, and accurately represents the topic's main research findings and methodologies.

Title: A Survey of Large Language Models
Table of Contents:

### E    Step by Step Method

Figure 5 illustrates one of the generation processes of the step-by-step generation method. Instead of generating the whole catalogue $C$ directly, we propose to generate step-by-step: $\mathbf{s}ource \rightarrow L_1 \rightarrow L_{1,2} \rightarrow L_{1,2,3}(\mathbf{t}arget)$, where $L_i$ is the $i$-th level headings of the catalogue. For example, in the first step, we input survey title and information about its reference documents and generate the first-level headings $L_1$ of the catalogue for that survey. Then, $L_1$ is added to the input, and the next step generates the first two levels headings $L_{1,2}$ of the

| | CEDS$_{sample}$ | |
|---|---|---|
| **Human**$_{sample}$ | Pearson r | 0.634 |
| | Pearson p | 0.027 |

Table 9: Consistency evaluation results between proposed metrics and corresponding human judgements.

catalogue. Finally, it generates the whole catalogue with a method similar to the previous step. The generation process corresponds to a decomposition of the conditional probability as:

$$P(\mathbf{t}|\mathbf{s}) = P(L_1|\mathbf{s})P(L_{1,2}|\mathbf{s}, L_1)P(L_{1,2,3}|\mathbf{s}, L_{1,2})$$

### F    Case Study

Table 10 and Figure 6 present an example of the best catalogues generated by end-to-end models and step-by-step models about the title "a survey of domain adaptation for neural machine translation".

Table 10 gives an example of alignment between catalogue items when calculating the Catalogue Edit Distance (CED). If an item does not match any other entity, then the required action for this item is insertion or deletion, so the cost is 1. If a node matches a node, then the required action is modification. The operation cost of modification is calculated according to the similarity. It is worth noting that Node[2] in the generated result does not match Node[1] in the ground truth, even though they have exactly the same content. It is because the former is a secondary heading, and the latter is a primary heading. If the alignment is forced, the cost required will be greater than the current result. Thus this example shows that not only does the metric CEDS measure catalogues from a semantic perspective, but also takes into account the hierarchical structure of the catalogue.

The parts marked in red in Figure 6 are the generation problems we observe. The first problem is duplicating the content in a single catalogue item, e.g. "monolingual monolingual data". The second one is the hierarchy error between sibling nodes. For example, "evaluation metrics" conceptually contain "automatic evaluation". Finally, there is a recurrence of the heading "applications of nmt domain adaptation". In summary, there are still duplication and hierarchical errors in the current results.

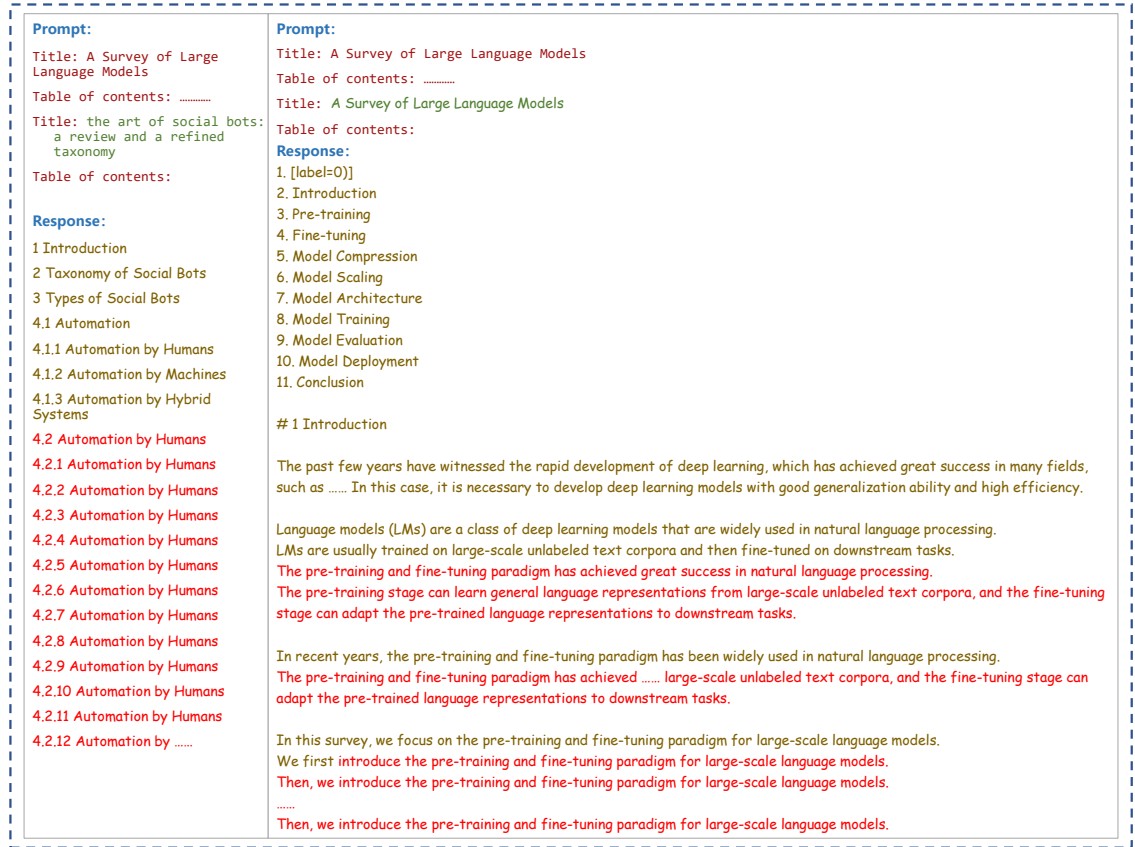

Figure 4: Two catalogue generation results of GAL model.

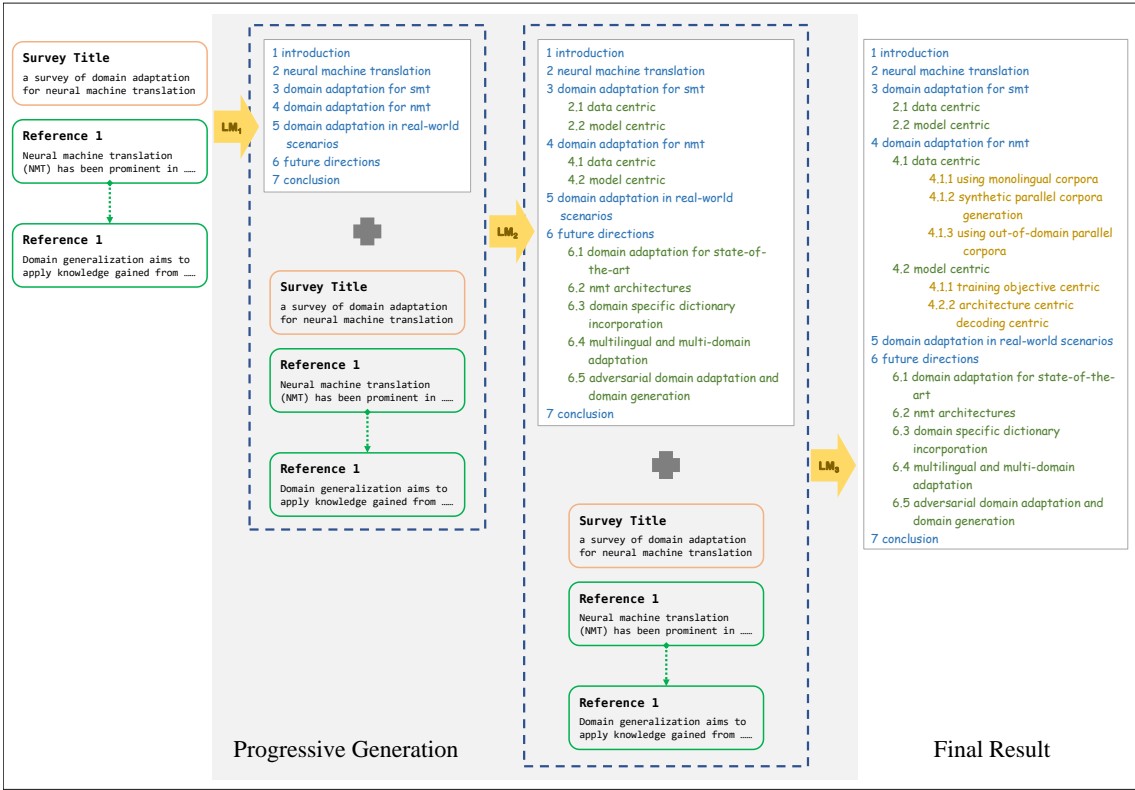

Figure 5: Architecture of the step-by-step approach.

Figure 6: Examples of end-to-end and step-by-step.

| Generated Result | Ground Truth | distance |
|---|---|---|
| $< l_1 >$ introduction [0] | $< l_1 >$ introduction [0] | 0.0 |
| - | $< l_1 >$ neural machine translation [1] | 1.0 |
| $< l_1 >$ background [1] | $< l_1 >$ domain adaptation for smt [2] | 0.50 |
| $< l_2 >$ neural machine translation [2] | $< l_2 >$ data centric [3] | 0.48 |
| $< l_2 >$ domain adaptation [3] | $< l_2 >$ model centric [4] | 0.35 |
| $< l_1 >$ survey of domain adaptation for nmt [4] | $< l_1 >$ domain adaptation for nmt [5] | 0.12 |
| $< l_2 >$ parallel data adaptation [5] | $< l_2 >$ data centric [6] | 0.34 |
| $< l_3 >$ fine-tuning [6] | $< l_3 >$ using monolingual corpora [7] | 0.44 |
| $< l_3 >$ instance weighting [7] | $< l_3 >$ synthetic parallel corpora generation [8] | 0.35 |
| $< l_3 >$ multi-task learning [8] | $< l_3 >$ using out-of-domain parallel corpora [9] | 0.38 |
| - | $< l_2 >$ model centric [10] | 1.0 |
| - | $< l_3 >$ training objective centric [11] | 1.0 |
| $< l_2 >$ monolingual data adaptation [9] | $< l_3 >$ architecture centric [12] | 0.44 |
| $< l_2 >$ synthetic parallel data adaptation [10] | $< l_3 >$ decoding centric [13] | 0.41 |
| - | $< l_1 >$ domain adaptation in real-world scenarios [14] | 1.0 |
| $< l_1 >$ conclusion and future directions [11] | $< l_1 >$ future directions [15] | 0.18 |
| - | $< l_2 >$ domain adaptation for state-of-the-art nmt architectures [16] | 1.0 |
| - | $< l_2 >$ domain specific dictionary incorporation [17] | 1.0 |
| - | $< l_2 >$ multilingual and multi-domain adaptation [18] | 1.0 |
| - | $< l_2 >$ adversarial domain adaptation and domain generation [19] | 1.0 |
| - | $< l_1 >$ conclusion [20] | 1.0 |

Table 10: An example of aligned nodes and distances between them when calculating CED.