# OpenReview forum: "Hierarchical Catalogue Generation for Literature Review: A Benchmark"
_EMNLP/2023/Conference — EMNLP 2023 Findings_

### Official Review · Reviewer_9uwx · 2023-08-04

**Soundness:** 3

**Excitement:**

3: Ambivalent: It has merits (e.g., it reports state-of-the-art results, the idea is nice), but there are key weaknesses (e.g., it describes incremental work), and it can significantly benefit from another round of revision. However, I won't object to accepting it if my co-reviewers champion it.

**Paper Topic And Main Contributions:**

This work proposes Hierarchical Catalogue Generation for Literature Review and builds a dataset for it. It also proposes two evaluation metrics for this task, i.e., Catalogue Edit Distance Similarity and Catalogue Quality Estimate.

**Questions For The Authors:**

1. How did you control the quaility of the constructed dataset?
2. How do you guarantee the evaluators are professional?
3. Will you consider build a human-annotated dataset and release it?

**Reasons To Accept:**

1. The proposed task is quite interesting.
2. The dataset could be helpful for future research in this filed.

**Reasons To Reject:**

1. This work is more suitable for the resource and evaluation track, as it does not focus on proposing new methods.
2. The reliability of the proposed metrics, i.e., Catalogue Edit Distance Similarity and Catalogue Quality Estimate, is not well verified. 450 samples are not enough.
3. The baselines in Sec. 5.1 are all before 2021.

**Reproducibility:**

3: Could reproduce the results with some difficulty. The settings of parameters are underspecified or subjectively determined; the training/evaluation data are not widely available.

**Reviewer Confidence:**

3: Pretty sure, but there's a chance I missed something. Although I have a good feel for this area in general, I did not carefully check the paper's details, e.g., the math, experimental design, or novelty.

---

> ### Author Rebuttal · Authors · 2023-08-28
>
> Thank you so much for your insightful comments.
>
> 1. **Track of paper**
>
> Thank you for your suggestion. Our initial idea is that hierarchical catalogue generation for literature review is a challenging task highly relevant to text summarization. We believe that our work can contribute a lot to the scientific summarization community.
>
> 2. **The reliability of proposed metrics and human evaluation**
>
> The reliability of the proposed metrics is assessed by rigorous  human evaluation. Survey catalogue generation is a challenging task, as it requires specialized background knowledge of relevant domains. Therefore, evaluating the survey catalogue costs a lot, which limits the sample size (450) of our evaluation. We invited three professional Ph.D. students who have published survey papers to make this evaluation reliable. The results of the three evaluators' assessments showed general statistical consistency with a Kappa coefficient of 0.28.
>
> 3. **Baselines before 2021?**
>
> We tested the performance of chatGPT on the catalog generation task.  Despite BART's earlier introduction, it remains a benchmark model for summarization tasks.
>
> 4. **Control the quality of the constructed dataset**
>
> In Section 2.2 (Dataset Construction), we provide an exhaustive explanation of our data collection process, including the decisions and compromises made to ensure optimal dataset quality.
>
> Initially, we identified 11,435 papers from arXiv, deemed suitable as literature review papers. PDF files of these papers are easy to collect but extracting catalogue from PDF files is difficult where structural information is usually dropped during converting. We finally choose to get the papers in HTML format available on the ar5iv website which processes articles from arXiv as responsive HTML web pages by converting from LaTeX via LaTeXML. However, we encountered a limitation where some authors had not uploaded their LaTeX code, resulting in a reduced dataset of 8,397 papers.
>
> Our final dataset, after eliminating data with fewer than five catalogue items and less than ten valid references, comprised 7,637 reference-catalogue pairs. To verify the accuracy of our extraction process, we manually inspected 100 randomly selected samples, confirming the correctness of all extracted catalogues.
>
> 5. **Human-annotated dataset**
>
> Large-scale data labeling is difficult and expensive to perform. We accept Review 2's comment and consider proposing a higher-level human-annotated test set  in the future.

---

### Official Review · Reviewer_bVQ3 · 2023-08-04

**Soundness:** 4

**Excitement:**

4: Strong: This paper deepens the understanding of some phenomenon or lowers the barriers to an existing research direction.

**Missing References:**

Surveytree: Automatic Generation of Survey Structures for NLP and AI topics

**Paper Topic And Main Contributions:**

This paper addresses the issue of lacking a clear and logical hierarchy in scientific literature reviews by proposing a task called Hierarchical Catalogue Generation for Literature Review. They construct a dataset and evaluation metrics, and benchmark different models to evaluate their capabilities in generating hierarchical catalogs for literature reviews.



**Questions For The Authors:**

see the reasons to reject

Besides,  what organization of surveys are good and what are not good do you think after analyzing the surveys you collected?

**Reasons To Accept:**

1. The research problem is well motivated.
2. The authors take in-depth analysis of the problem and propose new evaluation methods, which is quite innovative.
3. The authors construct a dataset to facilitate the future research.

**Reasons To Reject:**

1. Different people (or different research fields) have different styles of writing the literature review. So, before jumping into generating the survey catalogs, maybe analyzing what organizations are more common, or investigating what is the logic behind organizing a survey is important. (I think more investigation of the survey papers in the dataset is needed)

2. What are the main findings after the experiments (summarizing the reasons of outperformance of each model )?  I think this part is not clearly presented in the result analysis.

**Reproducibility:**

4: Could mostly reproduce the results, but there may be some variation because of sample variance or minor variations in their interpretation of the protocol or method.

**Reviewer Confidence:**

4: Quite sure. I tried to check the important points carefully. It's unlikely, though conceivable, that I missed something that should affect my ratings.

---

> ### Author Rebuttal · Authors · 2023-08-28
>
> Thank you so much for your insightful comments and afﬁrmation of this work.
>
> 1. **Organizations behind a survey**
>
> Intuitively, there are potential ways of organizations when writing a literature review. However, the underlying organization of writing requires in-depth knowledge of the specific field in order to mark it out, which is difficult to collect and analyze in large quantities.
>
> We observe a usually used organization style is "deduction and summary.” The definition of the topic to be synthesized is presented first. Then present the methods, data, and experiment results of the field. Advanced-level survey papers usually conduct their own experiments rather than simply citing data from other sources. The survey ends with the author's prospective outlook and reflections on the topic.
>
> The catalogue is the one that reveals how authors organize and can be collected quickly and in large quantities. Beyond standard section headings (Introduction, Background, Conclusion, References), the catalogue usually employs succinct, informative phrases to encapsulate chapter content.
>
> Although different people (or different research fields) have different styles, we collect review-catalogue pairs as much as possible and let the generative model learn by itself to implicitly derive the most appropriate organization. In the future, we plan to delve deeper into the organization and prior knowledge required for crafting literature reviews.
>
> We also plan to conduct a more comprehensive analysis of the quality of the survey papers in our dataset. This will include factors such as the authors' influence, the number of citations to the paper, and its publication status. Additionally, we're considering the incorporation of supplementary writing aids outside of the catalogue, such as human annotations and review paper labels, to assist in survey generation.
>
> These discussions and results will be integrated into our final version.
>
> 2.  **Main findings after experiments**
>
> (1) The hierarchical structure provided by a catalogue is crucial and beneficial in the process of generating literature reviews.
>
> (2) The task of generating a catalogue is indeed challenging, necessitating the generative models to possess an advanced comprehension of the references. Even for sophisticated large language models such as ChatGPT, this task remains a hard one.
>
> (3) End-to-end generation methods are superior to step-by-step generation methods based on BART-like models. There is still a lot of room for improvement in the future.
>
> 3. **Missing reference**
>
> We will cite this work in our final version: Surveytree: Automatic Generation of Survey Structures for NLP and AI topics.

---

### Official Review · Reviewer_SJpq · 2023-08-11

**Soundness:** 3

**Excitement:**

3: Ambivalent: It has merits (e.g., it reports state-of-the-art results, the idea is nice), but there are key weaknesses (e.g., it describes incremental work), and it can significantly benefit from another round of revision. However, I won't object to accepting it if my co-reviewers champion it.

**Paper Topic And Main Contributions:**

They present a task named Hierarchical Catalogue Generation for Literature Review as the first
step for review generation, which aims to produce a hierarchical catalogue of a review paper given various references. They construct a novel English Hierarchical Catalogues of Literature Reviews Dataset with 7.6k literature review catalogues and 389k reference papers. To accurately assess the model performance, They design two evaluation metrics for informativeness and similarity to ground truth from semantics and structure.
They get their papers from Arxiv and semantic scholars and extract the catalogues from HTML sources of the papers. They focus on surveys and review papers (appearance in the title) in the field of computer science.

**Questions For The Authors:**

1. Page 3, column 2, is ambiguous on the final size of the dataset after limiting it to computer-science only papers. is the final size 7637 or 4507?

**Reasons To Accept:**

1. nice study on the problem, and the claims,
2. proposing a new dataset for the task.

**Reasons To Reject:**

1. limited dataset size, and biased toward survey papers only, in computer science
2. very simple method, and idea. it doesn't make a remarkable improvement in the field.

**Reproducibility:**

3: Could reproduce the results with some difficulty. The settings of parameters are underspecified or subjectively determined; the training/evaluation data are not widely available.

**Reviewer Confidence:**

3: Pretty sure, but there's a chance I missed something. Although I have a good feel for this area in general, I did not carefully check the paper's details, e.g., the math, experimental design, or novelty.

---

> ### Author Rebuttal · Authors · 2023-08-28
>
> Thanks for your valuable suggestion.
>
> 1. **Dataset Size and Domain**
>
> Our dataset comprises a total of 7,637 reference-catalogue pairs, spanning a broad spectrum of domains (Computer Science, Mathematics, Physics, Electrical Engineering & Systems Science, Statistics, Quantitative Biology, Quantitative Finance, Economics).  In Table 6, we list the areas of papers included in the dataset and their corresponding quantities.
>
> With a majority share of 4,507 articles, Computer Science emerges as the most represented domain. Given the complexity of dataset annotation, which necessitates a high level of expertise to assess the quality of generation, we opt to conduct the catalogue generation task experiments within the most representative domain, i.e., Computer Science. Besides, surveys in other fields of the dataset provide a solid foundation and can further stimulate future research.
>
> A lot of effort has been put into collecting and processing the data (Section 2.2). Both Review 2 and 3 agree with our contribution. Besides, the size of our dataset exceeds that of other datasets related to surveys, as shown in the table below. Our data collection methodology is not only scalable but also adaptable to future expansion in terms of field and scale. In our final version, we will enhance the clarity of our expression.
>
> | Dataset | Size |
> | --- | --- |
> | HiCaD(Ours) | 7,637 |
> | BigSurvey-MDS[1] | 4,478 |
> | TutorialBank(survey)[2] | 390 |
> | Surveyor[3] | 297 |
> | Surfer100[4] | 100 |
> | DocStructure[5] | 27 |
>
> [1] Generating a Structured Summary of Numerous Academic Papers: Dataset and Method. (Liu et al., IJCAI 2022)
>
> [2] TutorialBank: A Manually-Collected Corpus for Prerequisite Chains, Survey Extraction and Resource Recommendation. (Fabbri et al., ACL 2018)
>
> [3] Surveyor: A System for Generating Coherent Survey Articles for Scientific Topics. (Jha et al., AAAI 2015)
>
> [4] Surfer100: Generating Surveys From Web Resources, Wikipedia-style. (Li et al., LREC 2022)
>
> [5] Document structure model for survey generation using neural network. (Xu et al., Frontiers of Computer Science)
>
> 2. **Method**
>
> The most important two contributions are as follows:
>
> (1) We observe that direct survey generation can lead to disorganized reviews with content repetition and logical confusion, an issue evident even in top-rated models like ChatGPT (Figure 1).
>
> (2) To enable the capability of generating reasonable hierarchical catalogues, we **are the first to** propose a novel and challenging task of Hierarchical Catalogue Generation for scientific Literature Review, which is far different from the previous tasks. We also propose a dataset as well as two evaluation metrics and adapt existing methods to the catalogue generation task.
>
> Note that, we devote our efforts in proposing a new task as well as a stable solution. We thoroughly compare extractive and abstractive approaches, end-to-end generation and step-by-step generation. In addition, we analyze the performance of large language models on this task. We believe our work can contribute to the development of the text summarization community.

---

### Meta-Review · Area_Chair_5CXw · 2023-09-18

**Recommendation:** 3

**Metareview:**

This paper presents a new task of generating a "catalogue" for literature review given reference papers. For this task, authors provide a dataset, evaluation measures, and evaluated performances of baseline models.
Reviewers agree on the novelty of the proposed task, and the value of the proposed dataset for further research in this field.
Reviewers also question the choice of baselines, data size, and the validity of the proposed measure, which are addressed by the authors. That being said, constructing a high-quality human-annotated test set is suggested.

---

### Decision · Program_Chairs · 2023-10-07

**Decision:**

Accept-Findings

**Comment:**

This paper presents a new task of generating a "catalogue" for literature review given reference papers. For this task, authors provide a dataset, evaluation measures, and evaluated performances of baseline models.
Reviewers agree on the novelty of the proposed task, and the value of the proposed dataset for further research in this field.
Reviewers also question the choice of baselines, data size, and the validity of the proposed measure, which are addressed by the authors. That being said, constructing a high-quality human-annotated test set is suggested.